# Physics-Aware Spatiotemporal Causal Graph Network for Forecasting with Limited Data

**Zijun Cui**[*]                                                   *cuizijun@msu.edu*
*Michigan State University*

**Sam Griesemer**[*]                                             *samgriesemer@usc.edu*
*University of Southern California*

**Sungyong Seo**[*]                                              *sungyongs@usc.edu*
*University of Southern California*

**Joshua Hikida**                                              *joshua.hikida@gmail.com*
*University of Southern California*

**Yan Liu**                                                     *yanliu.cs@usc.edu*
*University of Southern California*

**Reviewed on OpenReview:** *https://openreview.net/forum?id=n3yrVzPcNa*

## Abstract

Spatiotemporal models have drawn significant interest recently due to their widespread applicability across many domains. These models are often made more practically useful by incorporating beneficial inductive biases, such as laws or symmetries from domain-relevant physics equations. This "physics-awareness" provides an interpretable means of grounding otherwise purely data-driven models, improving robustness and boosting performance in settings with limited data. In this work, we view physical dynamics as domain knowledge that captures fundamental causal relationships across space and time, and can be effectively leveraged by our proposed physics-aware spatiotemporal causal graph network (P-STCGN). We firstly describe a means of deriving causal relationships from spatiotemporal data, serving as physics-aware labels to learn a causal structure via a dedicated neural module. We then formulate a forecasting module that can operate under this causal structure, producing predictions that are guided by physics-aware cause-effect relationships among modeled variables. Extensive experimentation demonstrates that our method is robust to noisy and limited data, outperforming existing models across a variety of challenging synthetic tasks and benchmark datasets. We further evaluate our method on real-world graph signals and observe superior forecasting performance, achieved by effectively utilizing causal signals from prior physics knowledge.

## 1 Introduction

Spatiotemporal modeling has drawn significant interest recently due to its wide application in climate, traffic systems, electricity networks, and many other fields. Complex machine learning models (e.g., deep neural networks) achieve superior performance in data-rich settings, such as computer vision, natural language processing, etc. Employing these models in spatiotemporal settings, however, poses a set of new challenges. The training data are often multi-resolution, of widely varying quality, and heavily constrained by physical principles. Further, real-world applications are typically characterized by a limited amount of training data. To address these challenges, integrating domain knowledge with data-driven models has emerged as one of the most promising directions forward, clearing a path for the construction of more robust and interpretable

pipelines. Specifically, physics-informed or physics-aware machine learning is of significant importance for spatiotemporal modeling as observations in the physical world (e.g., meteorological measurements and traffic flow) are naturally governed by physical principles.

Meanwhile, an emerging topic in deep learning community is causal inference and analysis in time series (Runge, 2018; Runge et al., 2019a; Nauta et al., 2019; Runge et al., 2019b). For example, Pamfil et al. (2020) introduce a smooth acyclicity constraint to multivariate time series inspired by (Zheng et al., 2018). Amortized causal discovery (ACD) considers an explicit separation between causal structure learning and the downstream dynamic prediction task (Löwe et al., 2022). While many works can discover causal structures directly from observational data, they often assume data sufficiency, requiring numerous samples for accurate causal discovery and being sensitive to data noise. Physics equations naturally connects to causality, rooted in the fundamental principle that *an effect can not occur before its cause.* Observations from the same dynamic system, even when driven by different underlying physics equations, share common causal relations across space and time that are intrinsically present in the dynamics.

For example, when the heat equation ($\frac{\partial u}{\partial t} = D\Delta u$ where $D$ is a diffusivity constant) is considered, we know that temporally first order and spatially second order derivatives are involved. We then specify causes and effects on a discrete domain (time interval $\Delta t$) as:

$$
\begin{aligned}
u_i(t+1) &= u_i(t) + \Delta t \cdot D\Delta u \\
&= u_i(t) + \Delta t \cdot D \sum_{j \in \mathcal{N}_i} (u_i(t) - u_j(t)),
\end{aligned}
\tag{1}
$$

where $\Delta$ is the Laplace operator and $\mathcal{N}_i$ is a set of adjacent nodes of $i$-th node. Eq. 1 shows the discrete Laplace operator. For the target value $u_i(t+1)$, the variables in the right-hand side are regarded as *known* causes from the heat equation. In the physical realm, the spatiotemporal observations, such as the climate and weather measurements (Kashinath et al., 2021), inherently adhere to physical principles. Physics equations are thus typically acknowledged as valuable information for robust spatiotemporal modeling. Nonetheless, the exploration of physics-aware causality remains scarce.

In this work, we introduce a novel physics-aware causal graph network (P-STCGN), unveiling physics-aware causality in dynamic systems. In our modeling process, we decouple causal structure learning from dynamic forecasting. Physics-aware causality is derived from prior physics knowledge. A causal module is introduced to learn causal relations from analytically derived physics-aware labels, such that it can capture the causal structures aligned with the physics laws. With the learned causal structure, a forecasting module learns hidden representations with corresponding *causes* to predict *effects*. Our main contributions are:

- We propose an innovative physics-aware causal structure learning approach. A causal module learns relationships given additional explicit labels extracted from physics knowledge.

- We present a novel physics-aware causal graph network that harnesses the insights derived from prior physics knowledge within a causal framework.

- We demonstrate the robustness of our physics-aware causal learning approach in handling noisy and limited data. It can be further extended to handle hidden confounders. Moreover, P-STCGN enhances forecasting performance over real-word graph signals and excels in terms of data efficiency and generalization.

## 2 Related work

**Physics-aware learning** Physics-informed learning is an emerging research direction where physics knowledge as strong inductive biases is utilized in the construction of interpretable and robust deep models (Meng et al., 2025). Different approaches have been proposed to incorporate physics knowledge into deep models, such as model architecture design. de Bezenac et al. (2018) showed how the design of a data-driven model can be motivated by the advection-diffusion equation to predict sea surface temperature. Seo et al. (2019)

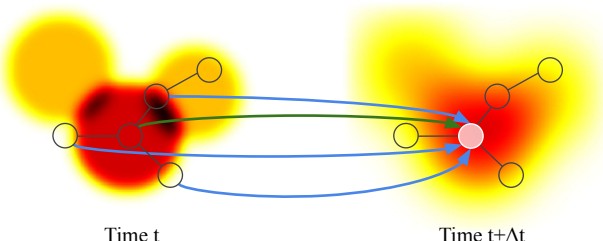

Figure 1: Heat dissipation over 2D space and time. Nodes in the graph structure correspond to sensors and the observations at each sensor are time varying. Given the heat equation ($\dot{u} = D\Delta u$), we can provide spatial (blue) and temporal (green) causal relations from previous nodes to a current target node (white).

constructed the proposed architecture inspired by a general form of PDEs in physical systems. Employing physics principles as training loss is another widely adopted approach, such as the Hamiltonian neural network (Greydanus et al., 2019). Physics knowledge has been used for spatiotemporal modeling since dynamic observations are usually governed by underlying dynamic mechanisms. Kaltenbach & Koutsourelakis (2021) propose a novel physics-aware generative state-space model for long-term predictions, and Wang et al. (2020) combine two turbulent flow simulation techniques with deep neural networks to predict physical fields. Despite these achievements, physics-aware causality is not very well studied in this context.

**Causal discovery in time series** Discovering underlying causal structure in time series data is a fundamental problem that remains actively studied today (Runge, 2018; Runge et al., 2019a; Nauta et al., 2019; Runge et al., 2019b). Rubin (1974); Pearl (2009); Imbens & Rubin (2015) introduce the problem and provide a mathematical framework for causal reasoning and inference under causal graphical models (also known as Bayesian networks (BN)) (Koller & Friedman, 2009). Granger (1969) formalizes a concept of quantifiable causality in time series, called Granger causality. Learning causal associations from time series is also an emerging topic in the deep learning community. Runge (2018) proposed a method to distinguish direct from indirect dependencies and common drivers among multiple time series to reconstruct a causal network. Runge et al. (2019b) quantify causal associations in nonlinear time series and Runge et al. (2019a); Nauta et al. (2019) provide promising applications of causal discovery in time series. Pamfil et al. (2020) introduce a smooth acyclicity constraint to multivariate time series inspired by (Zheng et al., 2018) who consider causal discovery a purely continuous optimization problem. Amortized causal discovery (ACD) is considered an explicit seperation between causal structure learning and the downstream dynamic prediction task (Löwe et al., 2022). Although many works are capable of discovering unknown causal structures from observational data directly, they usually assume data sufficiency, i.e., sufficient samples are available for accurate causal discovery. Besides, the performance can be sensitive to data noise. In contrast, we consider a physics-aware causal discovery approach. We leverage explicit causal relations inferred from domain knowledge as physics-aware causality for robust classification and retrieval under limited and noisy data.

**GNN-based spatiotemporal modeling.** While many graph-based architectures have been proposed to handle spatiotemporal observations on irregular domains (Li et al., 2018; Wu et al., 2020), one of the core issue behind graph-based neural networks (GNNs) (Hamilton et al., 2017; Kipf & Welling, 2017; Battaglia et al., 2018; Zhu et al., 2020) is that they cannot distinguish node-wise differences. To address this limitation, attention mechanisms have been extended to GNNs (Veličković et al., 2018; Zhang et al., 2018; Zheng et al., 2020). (Serrano & Smith, 2019), however, shows that attention weights do not necessarily correspond to importance for an output, but are instead more likely to be noisy predictors. Hence, there is a need for incorporation of *prior knowledge* into the unspecified representations between two nodes for reliable learning. Many graph-oriented methods leverage task-specific formulations (e.g., Zheng & Zhang (2023) for traffic, Zhang et al. (2020) for gesture recognition, etc) which can be effective, but sacrifice generality for well-tuned inductive biases. Physics-aware causality can be a direct solution to these challenges, being an effective way to dynamically incorporate problem-specific knowledge (e.g., spatiotemporal relations corresponding to arbitrary physics equations) into a general graph neural network architectural base.

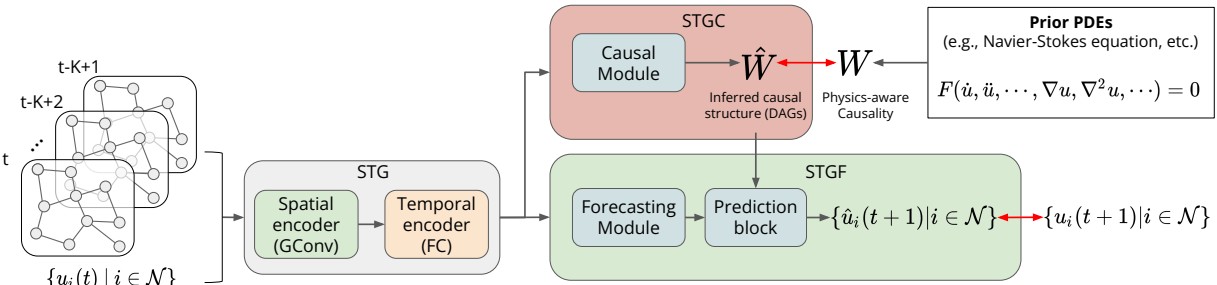

Figure 2: An overview of the proposed physics-aware spatiotemporal causal graph network (P-STCGN). A sequence of graph signals is firstly fed into a spatiotemporal graph network (STG), composed of a graph convolutional (GConv) spatial encoder and fully-connected (FC) temporal encoder. This is followed by two subsequent modules: (1) physics-aware causal module (STGC), and (2) spatiotemporal forecasting module (STGF). Prior PDEs from physical principles provide physics-aware causality. The red arrows denote how the supervised objectives are defined.

## 3 Problem formulation

Given observational data $\boldsymbol{X}_1, \cdots, \boldsymbol{X}_T$, where $\boldsymbol{X}_t \in \mathbb{R}^{N \times D_v}$, we assume that a static graph $\mathcal{G}_s = (\mathcal{V}_s, \mathcal{E}_s)$ is given (or can be constructed by features of each variable). The nodes $\mathcal{V}_s$ correspond to $\boldsymbol{X}_t \in \mathbb{R}^{N \times D_v}$ ($N$ nodes each associated with a $D_v$-dimensional observation vector) and the structure $\mathcal{E}_s$ is shared across different timestamps. With $N$ different nodes, the observations $\boldsymbol{X} \in \mathbb{R}^{T \times (N \times D_v)}$ can be regarded as a multivariate time series.

Additionally, we assume the existence of prior knowledge that could be beneficial for modeling the observations. This is a mild assumption since real-world observations are usually governed by physical principles, such as meteorological measurements obtained from sensors in an automatic weather station (AWS). For example, suppose a set of observations originate from weather sensors like those in an AWS: we expect domain-specific knowledge or equations related to weather phenomena to be beneficial for understanding the dynamics of the underlying weather system, and ultimately guide a learning method toward improved forecasting performance and generalizability. One particularly important prior equation for turbulent dynamics is the Navier-Stokes equation (Wang et al., 2020). These equations can be commonly represented as a function of spatial and time derivatives $F(\dot{u}, \ddot{u}, \cdots, \nabla u, \nabla^2 u, \cdots) = 0$, where $\dot{u}$ and $\ddot{u}$ denote the first and second-order time derivatives, respectively, and $\nabla$ represents the operator for the spatial derivative. As the continuous operators can be numerically decomposed in a discrete domain (e.g., onto graph structures), we can explicitly define *causes* for a target observation at time $t$ and extract causal relations accordingly. Note that causal relations derived from a particular equation are only partially complete due to the uncertainty surrounding the true governing equation. The available prior knowledge need only be partially relevant to the underlying dynamics in order to be beneficial.

Given the physics-aware causal relations, we can assign explicit labels between $NK$ variables, where $K$ is a maximum time lag for causality. In the length $K$ observations $\boldsymbol{X}_{t-K+1}, \cdots, \boldsymbol{X}_t$, there are $NK$ total mutually correlated observations, and we define $N_c$ causal relations among the $NK \times NK$ possible relations. In Fig. 1, we have $N = 5$ nodes in $\mathcal{G}_s$, and the total number of variables in the length $K = 2$ sequence is 10. Thus, there are 100 possible relations between the 10 variables, and the Heat equation (Eq. 1) elucidates $N_c = 13$ (5 temporal and 8 spatial) causal relations. We denote the causal graph $\mathcal{G}_c = (\mathcal{V}_c, \mathcal{E}_c)$ where $|\mathcal{V}_c| = NK$ and $|\mathcal{E}_c| = N_c$. Given the physics-aware causal relations ($\mathcal{G}_c$) derived from prior physics knowledge, our task is to find a model:

$$\hat{\boldsymbol{X}}_{t+1} = F(\boldsymbol{X}_{t-K+1}, \cdots, \boldsymbol{X}_t; \mathcal{G}_s, \mathcal{G}_c, \Theta), \tag{2}$$

where $\Theta$ is a set of learnable parameters in a model $F(\cdot)$.

# 4 Proposed model

We describe the details of our proposed model, namely Physics-aware Spatiotemporal Causal Graph Networks (P-STCGN). The P-STCGN employs a two-stage learning approach to explicitly decouple causal structure learning from dynamic forecasting. Fig. 2 shows a high-level view of P-STCGN consisting of two key modules: (1) physics-aware causal module (STGC), and (2) spatiotemporal forecasting module (STGF). STGC learns causal structure with causal labels derived from physics equations. Through STGC, we integrate the inductive bias into our model through semi-supervised causal structure learning. STGF then performs forecasting tasks using the learned causal structures. Both networks are designed to learn node representations from spatially and temporally correlated observations.

## 4.1 Physics-aware causality

Given an equation that is moderately beneficial for understanding the target dynamics, we can always define a causal graph given a physics equation, by decomposing the equation into causes and effects analytically. One particularly important prior equation for turbulent dynamics is the Navier-Stokes equation (Wang et al., 2020). Consider a fluid dynamics system governed by the Navier-Stokes equation. The causal graph $\mathcal{G}_c = <\mathcal{V}, \mathcal{E}>$ is defined as follows:

**Nodes ($\mathcal{V}$):** Each node $(i,j) \in \mathcal{V}$ represents a spatial location in the 2D fluid domain. The variables at each node include the velocity vector $\mathbf{u}_{i,j}(t)$ and pressure $p_{i,j}(t)$ at time $t$.

**Edges ($\mathcal{E}$):** The edges in the graph capture the causal relationships between nodes. Specifically, for each node $(i,j)$, the Navier-Stokes equation implies the following relationship for the velocity vector:

$$\mathbf{u}_{i,j}(t+1) = \mathbf{u}_{i,j}(t) - \frac{\Delta t}{\rho} \nabla p_{i,j}(t) + \Delta t \cdot \nu \nabla^2 \mathbf{u}_{i,j}(t) \tag{3}$$

Discretizing the spatial derivatives using central differences on a two-dimensional grid $(x, y)$, we get the following equation:

$$\nabla^2 \mathbf{u}_{i,j}(t) = \frac{1}{\Delta x \Delta y} \big( \mathbf{u}_{i+1,j}(t) + \mathbf{u}_{i-1,j}(t) + \mathbf{u}_{i,j+1}(t) + \mathbf{u}_{i,j-1}(t) - 4\mathbf{u}_{i,j}(t) \big) \tag{4}$$

where $\Delta x, \Delta y$ are the grid spacing.

This causal graph captures the dynamic interactions between different spatial locations in the fluid system according to the Navier-Stokes equation. We can also define the causal graph for the heat equation $\frac{\partial u}{\partial t} = D\Delta u$. These equations can be commonly represented as a function of spatial and time derivatives

$$F(\dot{u}, \ddot{u}, \cdots, \nabla u, \nabla^2 u, \cdots) = 0, \tag{5}$$

where $\dot{u}$ and $\ddot{u}$ denote the first and second-order time derivatives, respectively, and $\nabla$ represents the operator for the spatial derivative. Discretization of continuous operators (e.g., over graph structures) enables us to explicitly define *causes* for a target observation at time $t$ and extract causal relations accordingly. We further present a theorem to ensure the acyclicity of the defined causal graph given physics equations below. Its proof is included in Appx. A.

Consider a physics system described by a set of equations $\mathbb{E}$ that capture its dynamics. Let $\mathcal{G}_c = (\mathcal{V}, \mathcal{E})$ be the causal graph derived from these equations, where nodes $\mathcal{V}$ represent variables, and directed edges $\mathcal{E}$ represent causal relationships. If the physics equations in $\mathbb{E}$ are characterized by the following properties:

**(a) Linearity** The equations are linear, and no variable appears with nonlinear dependencies, expressed as

$$\frac{du_i}{dt} = \sum_j a_{ij} u_j + b_i, \quad \forall i$$

where $u_i$ is the i-th variable, $a_{ij}$ are coefficients, and $b_i$ represents external influences that are independent of $u$.

**(b) Causality preservation** Each variable's rate of change is solely determined by the values of variables in its causal neighborhood, including all immediate causes that affect the rate of change $u_i$, expressed as

$$\frac{du_i}{dt} = f_i(u_{\text{causal}}) + g_i(t),$$

where $u_{\text{causal}}$ represents the endogenous variables influencing $u_i$, and $g_i(t)$ represents explicitly modeled exogenous influences.

**(c) No instantaneous feedback** There are no instantaneous feedback loops where a variable directly influences itself in the same time step, expressed as

$$\frac{du_i}{dt} \neq h_i(u_i, t), \quad \forall i$$

Then, the causal graph $\mathcal{G}_c$ is acyclic.

These three properties (linearity, causality preservation, and the absence of instantaneous feedback) are commonly satisfied by physics equations. In particular, causality is inherently preserved in classical physics as well as in special and general theories of relativity (Riek & Chatterjee, 2021). The causal relations exist in the defined causal graph serve as physics-aware causal "labels" for our causal structure learning, as introduced in the following.

## 4.2 Architecture

We first learn node-wise latent representations by two modules: a spatial encoder (SE) and a temporal encoder (TE). Spatial encoders are designed to learn spatial dependencies at each timestamp via the static graph structure $\mathcal{G}_s$. The spatial encoder generates $K$ different snapshots which are grouped and fed into the temporal encoder as follows:

$$\{\boldsymbol{S}_{t'} = \text{SE}(\boldsymbol{X}_{t'}; \mathcal{G}_s) \mid t' = t - K + 1, \cdots, t\}, \tag{6}$$

$$\{\boldsymbol{Z}_{t'} = \text{TE}(\{\boldsymbol{S}_{t'-P}, \cdots \boldsymbol{S}_{t'}\}) \mid t' = t - K + 1, \cdots, t\}, \tag{7}$$

where $\boldsymbol{Z}_{t'} \in \mathbb{R}^{N \times D_c}$ is a set of node representations (dimension $D_c$) at time $t'$. $P$ is an aggregation order and TE merges the current embedding $\boldsymbol{S}_{t'}$ and past $P$ embeddings $\boldsymbol{S}_{t'-1}, \cdots, \boldsymbol{S}_{t'-P}$ for spatiotemporal node embeddings at $t'$. This temporal encoder does not consider the graph structure.

**Physics-aware causal module (STGC)** Once node embeddings are obtained, two $D_c$ dimensional vectors are fed into a causal module (CM), which computes a probability of association between the two corresponding nodes:

$$\hat{p}_{ji}^{t_j t_i} = \text{CM}(\boldsymbol{Z}_{t_j, j}, \boldsymbol{Z}_{t_i, i}), \tag{8}$$

where CM is a fully-connected network and $\boldsymbol{Z}_{t_j, j}$ is the $j$-th node's representation at time $t_j$. Since there are $N$ different nodes at each time $t$ (with a total of $K$ different timestamps), there are $N^2 K^2$ different settings for $p$. If observations are stationary and the causal relations are independent on the absolute timestamps $(t_j, t_i)$, but dependent on the relative time interval $\tau = t_i - t_j$, Eq. 8 can be reduced to $\hat{p}_{ji}^{\tau} = \text{CM}(\boldsymbol{Z}_{t_j, j}, \boldsymbol{Z}_{t_i, i})$.

It is worth noting that the causal module does not explicitly aim to recover some verifiably correct causal structure per se. Instead, it incorporates useful signal present in the graph $\mathcal{G}_c$, which itself represents causal links across time and space as determined by available governing physics equations. We therefore use the resulting "probability of association" $\hat{p}_{ji}^{t_j t_i}$ between two nodes as a *proxy* for a causal link, and references to outputs of the causal module hereafter should be taken as implicitly incorporating this fact (i.e., that the CM embodies a useful representation of a causal prior). This facilitates a more flexible means of integrating domain knowledge (compared to strictly enforcing some $\mathcal{G}_c$, say), as in practice, the equations from which the causal graph is defined may be incomplete or outright incorrect.

**Spatiotemporal forecasting module (STGF)** This module is used to learn node representations from spatiotemporal observations. It takes the learned causal structure from STGC and is used for the prediction of future signals. We introduce the forecasting module (FM) to transform the spatiotemporal representations $\boldsymbol{Z}$ to task-specific representations. As CM learns causality-associated representations, FM is adapted to learn prediction-associated representations.

$$\{\boldsymbol{H}_{t'} = \text{FM}(\boldsymbol{Z}_{t'}) \mid t' = t - K + 1, \cdots, t\}, \tag{9}$$

where $\boldsymbol{H}_{t'} \in \mathbb{R}^{N \times D_v}$. Since the causal relations from the $NK$ past variables to $N$ variables are inferred from STGC, the causal probabilities $\hat{p}_{ji}^{t_j t_i}$ (Eq. 8) are combined with $\boldsymbol{H}$ to predict next variables. Specifically, the output $\boldsymbol{H}$ from FM in STGF and $\hat{p}$ from CM in STGC are used to predict the next value at a node $i$ and time $t + 1$:

$$\hat{\boldsymbol{X}}_{t+1,i} = \sum_{t'=t-K+1}^{t-1} \sum_{j \in \mathcal{N}_i} \hat{p}_{j,i}^{t't} \cdot \boldsymbol{H}_{t',j}. \tag{10}$$

It's worth noting that we use causal probabilities between $t' \in [t-K+1, t-1]$ and $t$ instead of $t' \in [t-K+1, t]$ and $t + 1$. There are two reasons for this: (1) since $\boldsymbol{X}_{t+1}$ is not available, it is impossible to compute $\hat{p}^{t', t+1}$ (a function of $\boldsymbol{X}_{t+1}$) in advance, and (2) we assume that the causality is stationary and thus from $t'$ and $t$ is invariant if $\tau = t - t'$ is unchanged. The second assumption is particularly valid for spatiotemporal observations in physical systems as most of physics-based phenomena are not dependent on the absolute time but relative time intervals.

**Total Objective Function** The total loss function $\mathcal{L}$ consists of the causal structure loss $\mathcal{L}_c$ and the forecasting loss $\mathcal{L}_f$, as shown below:

$$\mathcal{L} = \lambda_f \mathcal{L}_f + \lambda_c \mathcal{L}_c = \lambda_f \sum_{i \in \mathcal{G}_s} \text{MSE}(\hat{\boldsymbol{X}}_{t+1,i}, \boldsymbol{X}_{t+1,i}) + \lambda_c \sum_{i,j \in \mathcal{G}_c} \text{CrossEntropy}(\hat{p}_{ji}^{t_j t_i}, \mathcal{E}_{ji}^{t_j t_i}) \tag{11}$$

where $\boldsymbol{X}_{t+1,i}$ is the ground truth observation, and $\mathcal{E}_{ji}^{t_j t_i}$ indicates the existence of an edge in the causal graph $\mathcal{G}_c$. The importance weights $\lambda_f = \lambda_c = 1$ when training the full model, but $\lambda_c$ is set to 0 when learning without the physics-aware causal structure while $\lambda_f$ is set to 0 when learning the causal structure alone.

**Non-causal labels for physics-aware causal structure learning.** In Section 3, we assume that the causal relations are given as explicit labels based on the prior equation (Eq. 5). The PDE provides information regarding which past and neighboring variables can be considered as possible *causes* for a current variable. It does not, however, provide information about which causal relations should be *excluded*. Since the physics-aware causal labels are highly imbalanced, CM will overfit on the positive-only labels. We address this challenge by introducing *non-causal* labels based on the principle that an effect can not occur before its cause. The non-causal labels are described as $\{n_{ji}^{t_j t_i} = 0 \mid t_i - t_j < 0\}$, capturing the set of relations where a timestamp $(t_j)$ of a candidate cause $(\boldsymbol{X}_{t_j,j})$ is later than that of a candidate effect $(\boldsymbol{X}_{t_i,i})$. Despite the availability of the non-causal labels, the imbalance issue still exists as the cardinality of $\{n_{ji}^{t_j t_i}\}$ is much larger than that of $\{p_{ji}^{t_j t_i}\}$. We mitigate this by subsampling the non-causal labels as many times as the available causal labels. More details about model configurations and training settings can be found in Appendix B.

## 5 Experimental results

We evaluate the proposed method in terms of both causal structure learning performance and dynamic forecasting performance. For causal structure learning, we evaluate the causal module (STGC) using synthetic and benchmark time series data. We further evaluate the performance with hidden confounding. For dynamic forecasting performance, we evaluate P-STCGN through a graph signal prediction task with real-world observations. Lastly, we discuss the scenarios where only partial prior equations are accessible.

### 5.1 Causal structure learning evaluation

Given $N$ different stationary series (or nodes), we train a model to predict if there exists significant temporal causal relationships between two time series: $\boldsymbol{X}_{t',j}$ and $\boldsymbol{X}_{t,i}$. Since the auto-regressive order is $P$, there

are potentially $NP \times N$ causal relations from $N$ variables $\boldsymbol{X}_{t'}$ where $t' \in [t - P, t - 1]$ to $N$ variables at time $t$. The true temporal causal relations are explicitly given as labels during training and a model is evaluated in two different aspects: (1) inter-causality classification and (2) intra-causality retrieval. For the *inter-causality classification*, we split a simulated multivariate time series into two parts across time axis: $\{\boldsymbol{X}_t | t = 1, \cdots, T_{train}\}$ and $\{\boldsymbol{X}_t | t = T_{train}, \cdots, T\}$. For the *intra-causality retrieval*, we only use a subset of the known labels to train a model and evaluate if it can retrieve the unseen labels correctly.

**Baselines** The task can be considered as *learning directional edge representations* from a variable at $t' \in [t - P, t - 1]$ to another variable at $t$, inspiring the three baselines as follows. First, we feed two node values into an MLP to predict the strength of causality. The other two baselines utilize a spatial and a temporal module to aggregate neighboring spatial/temporal values accordingly, after which the aggregated two node features are fed into an MLP to return the causal probability. For the spatial encoder (SE), we use GCN (Kipf & Welling, 2017), Chebyshev graph convolution networks (CHEB) (Defferrard et al., 2016), and GraphSAGE (Hamilton et al., 2017). The temporal encoder (TE) then concatenates node variables in the auto-regressive order. The STGC combines the two encoders spatiotemporally and the resultant node representations are fed into an MLP. Furthermore, we compare to causal discovery baselines: PARC (PCMCI (Runge et al., 2019b) based on partial correlations), Gaussian process regression and a distance correlation (GPDC), DYNOTEARS (Pamfil et al., 2020), and the SOTA Amortized Causal Discovery (ACD) (Löwe et al., 2022).

### 5.1.1 Synthetic study

We first generate multivariate time series $\boldsymbol{X} \in \mathbb{R}^{T \times N}$ from known temporal causal relations. Consider $N$ different stationary time series where each series influences the others in a time-lagged manner. At time $t$, a variable in $i$-th series $\boldsymbol{X}_{t,i} \in \mathbb{R}$ is defined as a function of variables at $t' < t$ such that $x_{i,t} = \sum_{t'=t-P}^{t-1} \sum_{j=1}^{N} f_{j,i}^{t',t}(\boldsymbol{X}_{t',j}) + \epsilon$ as described in (Runge et al., 2019b). $P$ is the auto-regressive order across time and $\epsilon$ is a noise term that is independent of other variable. Note that $f_{j,i}^{t',t}(\cdot)$ is regarded as a causal function from a previous variable at $(j, t')$ to a current variable $(i, t)$. Since the time series are stationary, the function $f_{j,i}^{t',t}(\cdot)$ can be relaxed as $f_{j,i}^{t-t'}(\cdot)$. We defined the temporal causal function in two different ways: (1) linear, and (2) non-linear conditional independence. For both settings, we generate length $T = 1000$ time series across $N = 7$ (linear) and $N = 13$ (non-linear) nodes. More details can be found in Appendix C.

**Inter-causality classification** The results on clean data (provided in Appendix C) demonstrate that the proposed model successfully outperforms other baselines on both settings. To further evaluate the robustness of the proposed model, we intentionally add i.i.d. noises to the generated time series. Since the time series are "contaminated" by the random noise after being causally generated, it becomes much more difficult to discover underlying temporal causality. Table 1 shows AUC of the models on the linear and non-linear settings. While AUCs are commonly decreased compared to the results on clean data, STGC can still learn meaningful representations from the spatiotemporal series unlike other methods. Note that when the scale of noise is increased ($\mathcal{N}(0, 5^2)$), MLP and spatial encoders followed by MLP are almost impossible to distinguish causal and non-causal relations (AUC is $\approx 0.5$), occurring also for TE+MLP for the non-linear series.

Furthermore, ACD requires a large amount of training data to produce accurate causal classification (its default training size is 10,000). In Table 1, the performance of ACD suffers from the limited training data; additive noise is generally a significant bottleneck for existing temporal causal discovery methods in multivariate time series settings. STGC, however, can learn robust representations for effective causal discovery. Supportive results are discussed in the Appendix C, comparing STGC against PARC, GPDC, and DYNOTEARS. Through comparisons, STGC is shown to be robust in causal classification with noisy and limited data by utilizing physics-aware causality.

**Intra-causality retrieval** We consider the time series generated from non-linear causality with added noise for evaluation. Note that there are 21 causal relations in the series that are split into two parts for training and testing. By adjusting the number of causal relations shown in training series, we can evaluate the robustness of the proposed model when the majority of causal relations are not given during the training process. Table 2 shows the average behavior when training on a subset of causal relations in time series.

Table 1: Inter-causality classification with additional noise.

| Model | Linear causality (AUC) | | Non-linear causality (AUC) | |
|---|---|---|---|---|
| | $\mathcal{N}(0, 1^2)$ | $\mathcal{N}(0, 5^2)$ | $\mathcal{N}(0, 1^2)$ | $\mathcal{N}(0, 5^2)$ |
| MLP | 0.611±0.029 | 0.506±0.010 | 0.517±0.013 | 0.499±0.004 |
| GCN+MLP | 0.507±0.004 | 0.500±0.001 | 0.502±0.002 | 0.500±0.004 |
| CHEB+MLP | 0.627±0.010 | 0.513±0.008 | 0.526±0.009 | 0.500±0.004 |
| SAGE+MLP | 0.621±0.021 | 0.516±0.006 | 0.527±0.007 | 0.502±0.003 |
| TE+MLP | 0.827±0.021 | 0.697±0.012 | 0.562±0.033 | 0.511±0.009 |
| ACD | 0.476±0.031 | 0.489±0.020 | 0.495±0.013 | 0.504±0.010 |
| STGC(Ours) | **0.849±0.020** | **0.712±0.013** | **0.640±0.012** | **0.582±0.007** |

Table 2: Intra-causality retrieval (AUC) from non-linear causal time series with $\mathcal{N}(0, 1^2)$

| Model | Number of train/test causality labels | | | |
|---|---|---|---|---|
| | 16 / 5 | 11 / 10 | 6 / 15 | 1 / 20 |
| TE+MLP | 0.550±0.031 | 0.546±0.023 | 0.539±0.028 | 0.501±0.011 |
| ACD | 0.486±0.030 | 0.506±0.014 | 0.499±0.012 | 0.500±0.016 |
| STGC (Ours) | **0.636±0.024** | **0.620±0.010** | **0.596±0.014** | **0.585±0.018** |

While TE+MLP detects some unseen causal relations when the number of labels shown for training is large (16), its performance quickly degrades as the number of available labels is decreased. STGC outperforms TE+MLP by a large margin, supporting the claim that STGC can extract more informative spatiotemporal representations. Interestingly, even if only a single causal relation is given as a known label (1/20), STGC still manages to retrieve unseen causal relations. Due to the lack of sufficient training samples, the ACD fails to perform effective causal discovery. Compared to ACD, STGC, by leveraging the physics-aware causality, is able to retrieve unseen causal relations.

### 5.1.2 Evaluation on benchmark datasets

Three benchmark datasets considered in the literature (e.g., Löwe et al. (2022)) are employed: Particles, Kuramoto (Kuramoto, 1975), and Netsim (Smith et al., 2011). Particles and Kuramoto are two fully-observed physics simulations. The Particles dataset simulates five moving particles in 2D, and some particles can influence others by pulling a spring. The Kuramoto dataset simulates five 1D time-series of phase-coupled oscillators. For both datasets, we follow the same settings as the synthetic study and generate $T = 1000$ time series for training. The Netsim dataset contains simulated fMRI data, and connectivity is defined between 15 regions of the brain. We follow the same settings as reported in (Löwe et al., 2022) and infer the connectivity across 50 samples.

Table 3 shows the results comparing STGC to ACD on the three benchmark datasets. From the results, we see that STGC significantly outperforms ACD. For example, on Kuramoto, P-STCGN achieves 40.6% accuracy improvement compared to ACD. On Particles, the performance of ACD is particularly poor due to the reduced data size. Though ACD achieves 0.999 AUC on the Particle dataset with sufficient data (10,000) as reported in (Löwe et al., 2022), its performance drops significantly with limited data (1,000). In contrast, we show that P-STCGN is capable of learning robust representations that aid in settings with limited data.

### 5.1.3 Hidden confounding

In this experiment, we study the effectiveness of the proposed approach with hidden confounding. We use the Particles dataset, where variation is introduced through an unobserved temperature variable. The temperature serves to modulate the strength of interactions among particles, where higher temperatures correspond to stronger forces and consequently, a more chaotic system. For each individual sample in the dataset, we

|  | Benchmark dataset | | |
| --- | --- | --- | --- |
| **Model** | Particles | Kuramoto | NetSim |
| ACD | 0.493 | 0.562 | 0.688 |
| STGC (Ours) | **0.520** | **0.968** | **0.925** |

Table 3: Comparison to ACD on benchmark datasets.

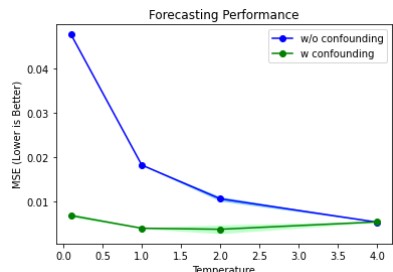

Figure 3: Forecasting with Hidden Confounding (MSE↓). Shaded regions represent the standard deviation across trials.

Table 4: Summary of results of prediction error (MSE) with standard deviations.

| | **TMAX** | | **TMIN** | |
| --- | --- | --- | --- | --- |
| **Model** | *Western* | *Eastern* | *Western* | *Eastern* |
| DCRNN | 0.1324±0.0024 | 0.1585±0.0033 | 0.0707±0.0017 | 0.1317±0.0028 |
| GCRN | 0.1336±0.0082 | 0.1588±0.0027 | **0.0701±0.0004** | 0.1302±0.0009 |
| FNO | 0.1234±0.0005 | 0.1963±0.0003 | 0.0906±0.0004 | 0.1676± 0.0001 |
| PA-DGN | 0.2620±0.0033 | 0.2921±0.0014 | 0.1720±0.0098 | 0.2346± 0.0009 |
| P-STCGN | **0.1111±0.0014** | **0.1355±0.0034** | 0.0731±0.0009 | **0.1262±0.0036** |
| | **SNOW** | | **PRCP** | |
| **Model** | *Western* | *Eastern* | *Western* | *Eastern* |
| DCRNN | 0.6757±0.0011 | 0.0406±0.0002 | 0.4703±0.0020 | 0.7588±0.0013 |
| GCRN | 0.6683±0.0012 | 0.0406±0.0001 | 0.4703±0.0009 | 0.7595±0.0001 |
| FNO | – | – | – | – |
| PA-DGN | 0.6626±0.0051 | 0.0402±0.0027 | 0.4979±0.0016 | 0.6819±0.0008 |
| P-STCGN | **0.6613±0.0035** | **0.0386±0.0007** | **0.4589±0.0033** | **0.6658±0.0025** |

draw an independent temperature $\mathcal{T}$ value from a categorical distribution $\mathcal{T} \sim \text{Categorical}([\alpha/2, \alpha, 2\alpha])$ following Löwe et al. (2022). This categorical distribution provides a probabilistic framework for assigning different temperature levels, allowing for the simulation of diverse scenarios within the particle system. In total, we consider four settings with categorical parameters $\alpha = \{0.1, 1, 2, 4\}$.

To model the temperature variable $\mathcal{T}$ as a hidden confounder, we introduce a hidden node in our model that affects the status of all particles. We also consider a baseline where no hidden confounder is assumed. We report both the structure learning accuracy (AUC) (Appendix Fig. 5) and the forecasting performance (MSE) (Fig. 3). As shown, including the hidden confounder in our model effectively achieves better causal structure discovery performance with higher AUC across different settings. In terms of forecasting performance, being able to model hidden confounders also achieves improved forecasting performance, especially with lower temperatures (i.e., $\alpha = \{0.1, 1, 2\}$).

## 5.2 Dynamic forecasting evaluation

To evaluate the dynamic forecasting performance of the proposed model, we consider a graph signal prediction task from real-world observations. The task is a prediction of future signals $\boldsymbol{X}_{t+1}$ given length $P = 10$ past spatiotemporal series $\boldsymbol{X}_{t-9} \cdots, \boldsymbol{X}_t$ under the graph structure.

Table 5: Data efficiency evaluation. 60% represents the full training set.

| Measurement | Model | % of available data used | | | |
|---|---|---|---|---|---|
| | | 5% | 10% | 20% | 60% |
| TMAX (West) | STGF | 0.1926±0.0937 | 0.1228±0.0014 | **0.1177±0.0029** | 0.1134±0.0003 |
| | P-STCGN | **0.1382±0.0034** | **0.1204±0.0005** | 0.1186±0.0007 | **0.1111±0.0014** |
| TMAX (East) | STGF | 0.1611±0.0047 | 0.1519±0.0039 | 0.1410±0.0039 | 0.1393±0.0011 |
| | P-STCGN | **0.1584±0.0043** | **0.1493±0.0022** | **0.1404±0.0013** | **0.1355±0.0034** |
| TMIN (West) | STGF | 0.1229±0.0120 | **0.0963±0.0063** | 0.0887±0.0040 | 0.0759±0.0024 |
| | P-STCGN | **0.1059±0.0080** | 0.0976±0.0012 | **0.0874±0.0014** | **0.0731±0.0009** |
| TMIN (East) | STGF | 0.1571±0.0020 | 0.1352±0.0112 | 0.1263±0.0116 | 0.1304±0.0038 |
| | P-STCGN | **0.1427±0.0047** | **0.1283±0.0026** | **0.1232±0.0035** | **0.1262±0.0036** |
| SNOW (West) | STGF | 1.3300±0.0685 | 0.9987±0.0100 | 0.8150±0.0208 | 0.6720±0.0070 |
| | P-STCGN | **1.2223±0.0051** | **0.9783±0.0076** | **0.7977±0.0051** | **0.6613±0.0035** |
| SNOW (East) | STGF | 0.0460±0.0014 | **0.0410±0.0003** | 0.0362±0.0003 | 0.0391±0.0008 |
| | P-STCGN | **0.0439±0.0005** | 0.0412±0.0083 | **0.0356±0.0001** | **0.0386±0.0007** |
| PRCP (West) | STGF | 0.5103±0.0042 | 0.4628±0.0020 | **0.4407±0.0024** | 0.4619±0.0047 |
| | P-STCGN | **0.5084±0.0012** | **0.4627±0.0014** | 0.4437±0.0022 | **0.4589±0.0033** |
| PRCP (East) | STGF | 0.8028±0.0029 | 0.8041±0.0060 | 0.7980±0.0151 | 0.6770±0.0042 |
| | P-STCGN | **0.7982±0.0029** | **0.7981±0.0029** | **0.7884±0.0012** | **0.6658±0.0025** |

**Dataset** We consider the climatology network[1] (Defferrard et al., 2020). Each sensor has 4 different daily measurements: TMAX: Maximum temperature (tenths of degrees C), TMIN: Minimum temperature (tenths of degrees C), SNOW: Snowfall (mm), and PRCP: Precipitation (tenths of mm). Each measurement is provided over 5 years from 2010 to 2014 (the length of series 1826), and we use them for our experiments. It is worth noting that the number of working sensors for each measurement is highly variable. While daily temperature observations are spatially densely available, the snowfall observations are comparatively sparse. Table 12 provides additional details for the dataset. The number of sensors from which the underlying graph was constructed is listed (along with the number of edges in the resulting graph). We split the series into training (60%), validation (10%), and testing (30%) sets. Additional details are in Appendix D.

**Baselines** We compare P-STCGN against two well-established data-driven baselines which have been introduced for similar tasks: DCRNN (Li et al., 2018) and GCRN (Seo et al., 2018). For physics-based baseline, we consider the PA-DGN (Seo et al., 2019) and FNO (Li et al., 2020). To adapt FNO for forecasting, we train it on observational data using its default settings and test it for a future time step prediction.

**Causality labels from PDEs** There are no ground truth PDEs for this dataset. We thus consider the PDEs among the family of the continuity equation, e.g., Navier-Stokes equations. These equations commonly describe how target observations are spatiotemporally varying with respect to its second-order spatial derivatives and first-order time derivative. In the underlying graph structure, spatially 1-hop neighboring nodes ($j \in \mathcal{N}_i$) are considered as adjacent causes to the observation at the $i^{\text{th}}$ node, and observations at $t-1$ are potential causes to the observations at $t$ autoregressively. The existing causal labels can be described as $\{p_{ji}^{t_j t_i} = 1 \mid t_i - t_j = 1 \text{ and } j \in \mathcal{N}_i\}$.

### 5.2.1 Prediction accuracy

We use mean squared error (MSE) as a metric to compare P-STCGN against the external baselines[2]. Table 4 shows that P-STCGN mostly outperforms other baselines across different regions and measurements. Both

---

[1]Global Historical Climatology Network (GHCN) provided by National Oceanic and Atmospheric Administration (NOAA).
[2]For SNOW and PRCP, FNO was unable to converge during training, likely due to the spatially sparse sensors with discrete measures.

Table 6: Forecasting with partially available prior (MSE)

| # equations withheld | % training data | |
|:---:|:---:|:---:|
| | 100% | 10% |
| 0 | 3.28±0.17 | 4.71±0.37 |
| 2 | 3.27±0.18 | 4.72±0.35 |
| 4 | 3.36±0.16 | 4.86±0.49 |
| 5 | 3.30±0.19 | 4.99±0.46 |

DCRNN and GCRN replace fully connected layers in the RNN variants (GRU and LSTM) with diffusion convolution and Chebyshev convolution layers. Thus, they similarly aggregate spatiotemporal correlation, exemplified by the close prediction error. Compared to the data-driven DCRNN and GCRN, our approach achieves better accuracy, particularly for TMAX and PRCP. For example, P-STCGN improves DCRNN by 16% for TMAX (western) prediction. Compared to two physics-based baselines PA-DGN and FNO, we also observe performance improvements. Particularly, P-STCGN improves PA-DGN significantly for TMAX and TMIN, demonstrating that our means of incorporating prior physics knowledge is much more effective.

### 5.2.2 Ablation study

To further study the effectiveness of P-STCGN, we perform ablation studies on data efficiency and generalization ability. We compare its performance to the baseline model STGF (i.e., just the forecasting base without explicitly incorporating physics-informed causal signal).

**Data efficiency** We consider different training sets with reduced number of training samples of 5%, 10%, and 20%. In comparison, we also consider the full training set (60%). Results are shown in Table 5. Compared to STGF, we can clearly see how the additional physics-aware causality is beneficial for modeling spatiotemporal data, particularly on extremely limited data (5%). Specifically, P-STCGN improves STGF by 28% on TMAX (western), implying that the PDE-based causal labels are significantly informative and can help compensate for the lack of training samples. As the training set size increases, the MSE difference between P-STCGN and STGF reduces. Nevertheless, P-STCGN continues to outperform STGF by a decent margin.

**Generalization Ability** We also consider the generalization ability, and as a proxy we train P-STCGN and STGF on one region and test on another. We observe that P-STCGN consistently outperforms STGF in most scenarios. Detailed results are reported in Appendix D.

### 5.2.3 Interpretation of learned causality

Once the causal module is trained based on a prior PDE, we can use it to examine how potential causes cary over space and time. We visualize how the causal probability is changed in Appendix D. From the visualizations, we can observe that variables spatially close to current observations have higher causal association for PRCP and TMAX. Additionally, we can infer the strength of causal relations between neighboring sensors and a specified target sensor (as shown in Appendix D). As shown, the physics-aware causality is not only informative for spatiotemporal modeling but also enables the discovery of unspecified causal relations.

## 5.3 Discussions on partially available prior

We investigate the effectiveness of utilizing partially available physics priors to enhance forecasting performance, addressing a common challenge in real-world applications where complete prior knowledge is unavailable. Employing the synthetic experiment settings detailed in Section 5.1.1, we utilize synthetic data for forecasting evaluation, focusing on the linear case with 7 equations. To evaluate, we present forecasting performance under different scenarios, varying the number of equations withheld. For comparison, we report performance with both 100% training data and only 10% training data. The results are summarized in Table 6. The

observed trends indicate that the presence of a physics prior, even when partially available, contributes to improved forecasting accuracy, particularly in low-data settings.

## 6 Conclusion

In this paper, we introduced a novel physics-aware spatiotemporal causal graph network (P-STCGN). This approach enables dynamic forecasting alongside a learned causal structure, capturing relevant information from physics domain knowledge. We evaluated the proposed framework from two primary perspectives: inter-causality classification and intra-causality retrieval. Our experimental results highlight the effectiveness of the physics-aware causal learning approach compared to common alternatives, especially in scenarios involving noisy and limited data. We further evaluated the forecasting performance on real-world observations from climate systems, and we observed superior accuracy due to our method utilizing physics-informed causal relations, even when the physics priors are inaccurate.

## Acknowledgements

This work was supported in part by the National Science Foundation under awards #2226087 and #1837131, in addition to the USC Ershaghi Center for Energy Transition (E-CET).

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

## A    Physics-aware Causality

**Proof of Theorem 1:** Assume that the physics equations in $\mathbb{E}$ satisfy the properties of linearity, causality preservation and the absence of instantaneous feedback loops. Define the construction of the causal graph $\mathcal{G}_c = (\mathcal{V}, \mathcal{E})$ based on the physics equations. Let each variable correspond to a node, and each directed edge represent a causal relationship. In linear systems, each variable's rate of change is a linear combination of its influencing variables. According to the causality preservation, each variable's rate of change is solely determined by the values of variables in its causal neighborhood. Hence, there are directional dependencies from influencing variables to influenced variables. Both linearity and causality preservation prevents circular dependencies, ensuring that there are no loops formed among variables through the causal relationships derived based on physics equations. No instantaneous feedback property further eliminates the possibility of loops involving instantaneous feedback in the causal graph $\mathcal{G}_c$. Combining three properties, we can conclude that each variable's rate of change is dictated by a linear combination of its immediate causes, and there are no circular dependencies introduced. Therefore, the derived causal graph is acyclic.

## B    Model configuration and experimental details

### B.1    Model configuration

STG consists of two parts: SE and TE, and STG is shared in STGC and STGF. SE is defined by two-layer of GraphSAGE with 32 hidden units. TE concatenates the output of SE in a temporal axis, $[\boldsymbol{X}_{t-P}; \cdots ; \boldsymbol{X}_t]$. CM is an MLP [FC32,ReLU,FC1,Sigmoid], and VM is an MLP [FC32,ReLU,FC32,ReLU,FC1] where FC($n$) denotes a fully-connected layer with $n$ units. The baselines are defined to have a similar number of learnable parameters to P-STCGN.

### B.2    Training settings

We train P-STCGN for all tasks with a batch size of 32 on a single GPU (NVIDIA T4 GPU) for 1000 epochs with early stopping (ending early when the validation error is not improved for 20 epochs). All results in the paper are mean values from 10 different random seeds.

In Table 7, we report training times for both STGF (no causal module) and P-STCGN (with causal module) on the dynamic forecasting task using the climatology network presented in Section 5.2. Reported times are averaged across 10 trials on the above mentioned hardware for 1000 training epochs over the respective time series (TMAX, TMIN, SNOW, PRCP). Note that the number of available samples for each series varies; see Table 12 for additional dataset details. Inference times were also measured post-training, averaging to 3.5s per 100k output values across all settings. Inference times are approximately uniform given the underlying model architecture is the same for each setting. Table 8 reports the number of parameters in each of the model components used for this task. Taken together, these tables provide a cohesive view of the dataset size, model size, and computational cost required to reach the reported performance, serving as a loose guide for understanding the scale of model deployment on other real-world tasks.

## C    Causal structure learning evaluation

### C.1    Synthetic time series data generation

**Data generation**    We first generate multivariate time series $\boldsymbol{X} \in \mathbb{R}^{T \times N}$ from known temporal causal relations. Consider $N$ different stationary time series where each series influences the others in a time-lagged manner. At time $t$, a variable in the $i$-th time series $\boldsymbol{X}_{t,i} \in \mathbb{R}$ is defined as a function of variables at $t' < t$ such that:

$$x_{i,t} = \sum_{t'=t-P}^{t-1} \sum_{j=1}^{N} f_{j,i}^{t',t}(\boldsymbol{X}_{t',j}) + \epsilon, \tag{12}$$

Table 7: Observed training times over the 1,000 epoch horizons, averaged across the 10 trials reported for the climatology task in the main paper.

| Model | TMAX | | TMIN | |
| | *Western* | *Eastern* | *Western* | *Eastern* |
|---|---|---|---|---|
| STGF | 5974.93s±117.02s | 3520.79s±31.18s | 6124.94s±88.19s | 3517.35s±79.61s |
| P-STCGN | 6673.39s±53.47s | 3872.72s±41.84s | 7247.38s±99.86s | 3919.36s±150.29s |

| Model | SNOW | | PRCP | |
| | *Western* | *Eastern* | *Western* | *Eastern* |
|---|---|---|---|---|
| STGF | 678.63s±1.64s | 1711.29s±3.34s | 4710.77s±96.42s | 4678.70s±146.29s |
| P-STCGN | 764.53s±1.36s | 1878.54s±6.78s | 5638.68s±54.97s | 4942.19s±15.97s |

Table 8: Parameter counts for model components used on the climatology forecasting task.

| Model component | # of parameters |
|---|---|
| Encoders (spatial/temporal) | 2,176 |
| Causal module | 22,593 |
| Value module | 12,385 |
| STGF | 14,561 |
| P-STCGN | 37,154 |

as described in [3] (Runge et al., 2019b). Further breaking down this formulation seen in Section 4.1, we generate two series based on linear and nonlinear causality:

**Linear causality**

$$\boldsymbol{X}_{t,0} = 0.7\boldsymbol{X}_{t-1,0} + \epsilon$$
$$\boldsymbol{X}_{t,1} = 0.8\boldsymbol{X}_{t-1,1} + 0.8\boldsymbol{X}_{t-1,3} + \epsilon$$
$$\boldsymbol{X}_{t,2} = 0.5\boldsymbol{X}_{t-1,2} + 0.5\boldsymbol{X}_{t-2,1} + 0.6\boldsymbol{X}_{t-3,3} + \epsilon$$
$$\boldsymbol{X}_{t,3} = 0.4\boldsymbol{X}_{t-1,3} + \epsilon$$
$$\boldsymbol{X}_{t,4} = 0.9\boldsymbol{X}_{t-2,2} + 0.1\boldsymbol{X}_{t-3,6} + \epsilon$$
$$\boldsymbol{X}_{t,5} = 0.2\boldsymbol{X}_{t-1,0} + 0.2\boldsymbol{X}_{t-2,0} + 0.2\boldsymbol{X}_{t-3,0} + \epsilon$$
$$\boldsymbol{X}_{t,6} = \epsilon$$

where $\epsilon \sim \mathcal{N}(0,1)$. In the linear causal series, there are 12 causal relations between $N = 7$ series and the maximum time lag in the causal relations is 3.

---

[3]https://github.com/jakobrunge/tigramite

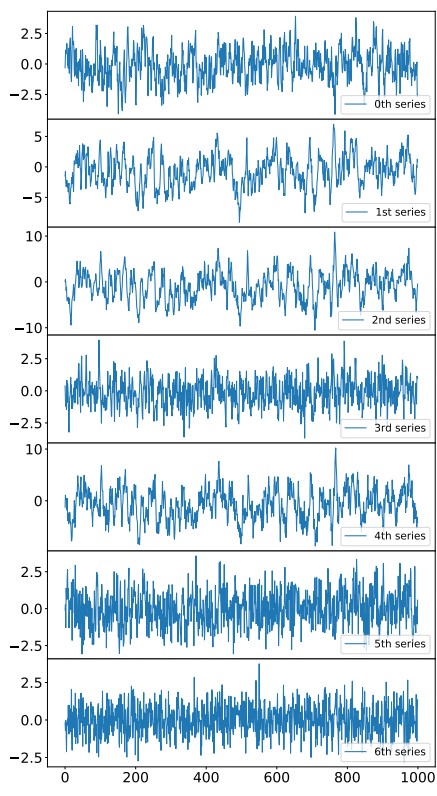

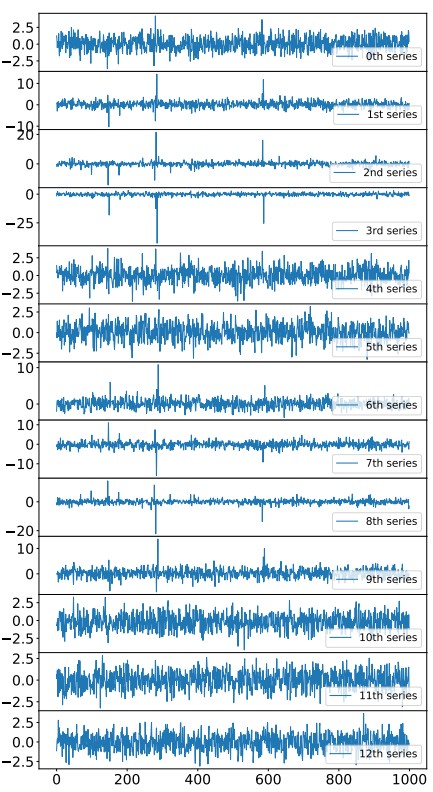

(a) Generated multivariate time series from given linear causal relations.

(b) Generated multivariate time series from given nonlinear causal relations.

**Nonlinear causality**

$$\boldsymbol{X}_{t,0} = \epsilon$$
$$\boldsymbol{X}_{t,1} = 0.2(\boldsymbol{X}_{t-1,1})^2 + 0.7\boldsymbol{X}_{t-2,2} + \epsilon$$
$$\boldsymbol{X}_{t,2} = 0.3(\boldsymbol{X}_{t-2,0})^3 + 0.05\boldsymbol{X}_{t-1,3} + \epsilon$$
$$\boldsymbol{X}_{t,3} = -0.09(\boldsymbol{X}_{t-3,2})^2 + 0.4\boldsymbol{X}_{t-1,5} + \epsilon$$
$$\boldsymbol{X}_{t,4} = 0.2(\boldsymbol{X}_{t-1,0})^2 + 0.01\boldsymbol{X}_{t-3,1} - 0.2(\boldsymbol{X}_{t-1,5})^2 + \epsilon$$
$$\boldsymbol{X}_{t,5} = \epsilon$$
$$\boldsymbol{X}_{t,6} = 0.3\boldsymbol{X}_{t-1,5} + 0.3\boldsymbol{X}_{t-2,4} - 0.3\boldsymbol{X}_{t-3,3} + \epsilon$$
$$\boldsymbol{X}_{t,7} = -0.2(\boldsymbol{X}_{t-1,0})^2 + 0.7\boldsymbol{X}_{t-2,8} + \epsilon$$
$$\boldsymbol{X}_{t,8} = -0.3(\boldsymbol{X}_{t-1,0})^3 + 0.05\boldsymbol{X}_{t-2,0} + \epsilon$$
$$\boldsymbol{X}_{t,9} = 0.9\boldsymbol{X}_{t-3,1} + \epsilon$$
$$\boldsymbol{X}_{t,10} = -0.02\boldsymbol{X}_{t-1,0} + 0.1\boldsymbol{X}_{t-3,6} - 0.2(\boldsymbol{X}_{t-1,4})^2 + \epsilon$$
$$\boldsymbol{X}_{t,11} = -0.3\boldsymbol{X}_{t-4,0} + \epsilon$$
$$\boldsymbol{X}_{t,12} = -0.3\boldsymbol{X}_{t-1,11} + \epsilon$$

where $\epsilon \sim \mathcal{N}(0,1)$. In the nonlinear causal series, there are 22 causal relations between $N = 13$ series and the maximum time lag in the causal relations is 4 (See $\boldsymbol{X}_{t,11}$). When we conduct the intra-causality retrieval experiment, we feed length 4 series from $\boldsymbol{X}_{t-3}$ to $\boldsymbol{X}_t$ to the classifier. Thus, the causality from time lag 4 in $\boldsymbol{X}_{t,11}$ is not labelled. Figure 4b shows generated sample time series based on the formulation above.

**Data preprocessing** Given $N$ different generated time series (length $T = 1000$), we assume a fully connected graph structure between the multivariate time series. We use the first 50% of the series data for

Table 9: Inter-causality classification (clean data)

| Model | Linear causality | | |
| | Recall | AUC | CE |
|---|---|---|---|
| MLP | 0.579±0.124 | 0.670±0.012 | 0.611±0.015 |
| GCN+MLP | 0.193±0.126 | 0.508±0.008 | 0.669±0.004 |
| CHEB+MLP | 0.577±0.055 | 0.677±0.010 | 0.585±0.017 |
| SAGE+MLP | 0.554±0.161 | 0.668±0.035 | 0.583±0.014 |
| TE+MLP | 0.756±0.038 | 0.858±0.020 | 0.435±0.026 |
| STGC | **0.767±0.023** | **0.885±0.011** | **0.340±0.035** |
| Model | Non-Linear causality | | |
| | Recall | AUC | CE |
| MLP | 0.365±0.211 | 0.533±0.023 | 0.658±0.013 |
| GCN+MLP | 0.241±0.194 | 0.511±0.002 | 0.677±0.013 |
| CHEB+MLP | 0.416±0.124 | 0.551±0.013 | 0.650±0.011 |
| SAGE+MLP | 0.367±0.101 | 0.554±0.006 | 0.637±0.015 |
| TE+MLP | 0.438±0.107 | 0.611±0.051 | 0.625±0.017 |
| STGC | **0.503±0.041** | **0.689±0.013** | **0.522±0.015** |

the training series, and the following 20% for the validation series. The remaining 30% of the series is used to evaluate the baselines and P-STCGN.

## C.2  Ablation study on clean synthetic data

For the inter-causality classification task, we report the mean of recall, AUC, and cross entropy error (with standard deviation) on the test series. In this task, the temporal causality among the potential relations ($NP \times N$) is sparse, implying the recall, which tells how many actual causal relations are retrieved, is particularly important. We evaluate the proposed model on two different settings: (1) linear, and (2) non-linear temporal causality. The results in Table 9 demonstrate that the proposed model successfully outperforms other baselines on both settings. More specifically, all models are able to distinguish non-causal and causal relations in the linear setting according to AUC. However, the temporal change is particularly important to understand the causality among the variables. For the non-linear setting, the results show that all metrics from models are degraded significantly compared to the linear setting. Nonetheless, the temporal information is more important but the spatial information can still be helpful (STGC vs. TE+MLP).

## C.3  Comparison to causal discovery methods

Table 11 shows recall from existing causal discovery in multivariate time series methods (PCMCI based on partial correlations (PARC) and Gaussian process regression and a distance correlation (GPDC)) on the non-linear series. It shows that STGC is able to learn robust representations for the causal discovery from noisy series by utilizing the explicitly given labels.

# D  Forecasting evaluation

## D.1  Real-world graph signal data

**Data preprocessing**   We first sample time series from the entire sensor array to construct more localized graph signals. The number of available sensors is dependent on the type of measurement. Given multivariate time series from multiple sensors, we construct a distance-based graph structure using a $k-$NN algorithm where $k = 2$. The value of $k$ is chosen such that graph density is properly balanced, and to ensure that a

Table 10: Generalization evaluation across the two regions.

| | TMAX | | | |
|---|---|---|---|---|
| | Western | | Eastern | |
| **Model** | *Train on Western* | *Train on Eastern* | *Train on Eastern* | *Train on Western* |
| STGF | 0.1134±0.0014 | 0.1256±0.0027 | 0.1393±0.0011 | 0.1556±0.0024 |
| P-STCGN | **0.1111±0.0014** | **0.1240±0.0015** | **0.1355±0.0034** | **0.1532±0.0019** |
| | TMIN | | | |
| | Western | | Eastern | |
| **Model** | *Train on Western* | *Train on Eastern* | *Train on Eastern* | *Train on Western* |
| STGF | 0.0759±0.0024 | **0.0906±0.0040** | 0.1304±0.0038 | 0.1308±0.0028 |
| P-STCGN | **0.0731±0.0009** | 0.0919±0.0021 | **0.1262±0.0036** | **0.1284±0.0014** |

Table 11: Recall for causal discovery methods

| **Noise** | PARC | GPDC | DYNOTEARS | STGC |
|---|---|---|---|---|
| $\mathcal{N}(0, 1^2)$ | 0.48 | 0.48 | 0.29 | 0.66 |
| $\mathcal{N}(0, 5^2)$ | 0.00 | 0.00 | 0.00 | 0.48 |

sensor is only connected with other spatially close sensors. It is worth noting that the number of working sensors for each measurement is highly variable. While daily temperature observations are spatially densely available, the snowfall observations are comparatively sparse. Table 12 provides additional details for the dataset. The number of sensors from which the underlying graph was constructed is listed (along with the number of edges in the resulting graph).

Each observation is multiplied by a scalar (0.01) to be normalized and provide numerically stable computation. We used the first 60% of the series data for the training set, and the following 10% for the validation series. The remaining 30% series is used to evaluate the baselines and P-STCGN.

### D.2 Ablation ability on graph signal prediction

**Generalization ability.** To further study the effectiveness of incorporating the physics-aware causality, we study the generalization ability of P-STCGN. In particular, we perform an ablation study where we train P-STCGN and STGF on one region and test on another region. We consider the TMAX and TMIN for evaluation and the results are reported in Table 10.

### D.3 Interpretation of learned causality

Once the causal module is trained based on a guiding PDE, we can use the module to examine how the potential causes are varying over space and time. In the following, we show how the causal probability is changed on the two regions. P-STCGN extracts causality-associated information from spatiotemporal series. In Fig. 6a, we can see that variables spatially close to current observations have higher causal assocation for PRCP and TMAX. A similar pattern appears on another region (shown in Fig. 6b). On the other hand, sensors for SNOW are more related to sensors farther away.

We can also infer which neighboring sensors have stronger/weaker causal relations to a specified target sensor. In Fig. 6c and 6d, 4 sensors are sampled from each region to visualize how much their $K-$hop neighboring variables are causally related. We can see that daily max temperatures from the sensor 42 in the western region have been strongly affected by spatially close (2-hop) sensors, however, the max temperature at the

Table 12: Information on sensor networks from the climate dataset.

| **Western** | TMAX | TMIN | SNOW | PRCP |
|---|---|---|---|---|
| # of sensors | 434 | 423 | 31 | 319 |
| # of edges | 1142 | 1110 | 76 | 862 |
| **Eastern** | TMAX | TMIN | SNOW | PRCP |
| # of sensors | 244 | 248 | 114 | 323 |
| # of edges | 632 | 636 | 298 | 844 |

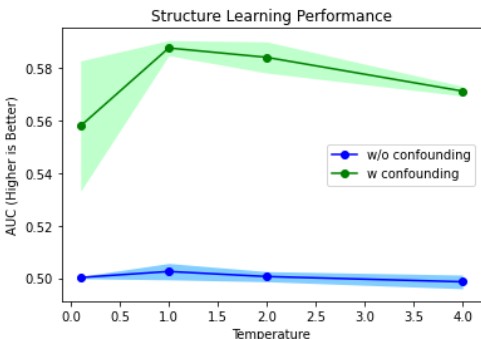

Figure 5: Causal discovery with Hidden Confounding (AUC↑). Shaded regions represent standard deviations.

sensor 25 is more likely dependent on sensors a bit far away (6 or 7-hop). On the other hand, sensor 26 is more dependent on mid-range sensors (4 or 5-hop). In eastern states, sensors 2 and 3 are associated with closer sensors; however, sensor 0 and 1 do not have distinct causal relations from their neighboring sensors. We find that the physics-aware causality is not only informative for spatiotemporal modeling directly but also enables the discovery of unspecified causal relations.

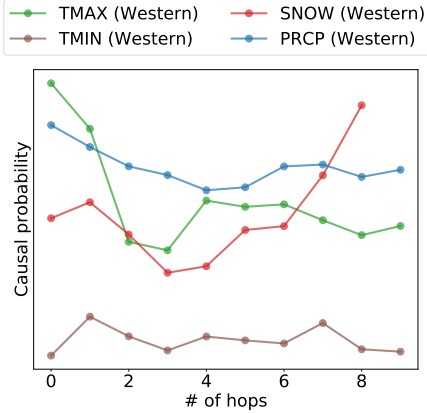

(a) Average causal probability curves vs. the number of hops over all sensors in Western region.

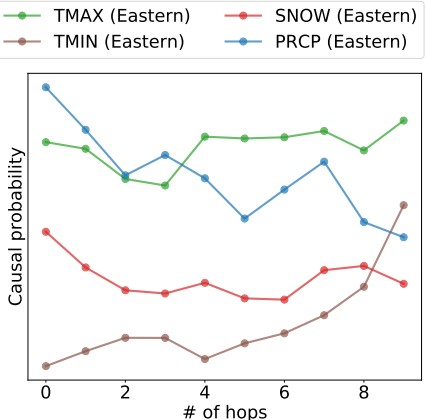

(b) Average causal probability curves vs. the number of hops over all sensors in Eastern region.

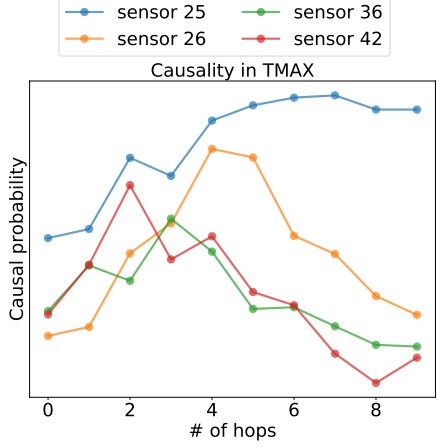

(c) Average causal probability curves vs. the number of hops from particular sensors in Western region (TMAX).

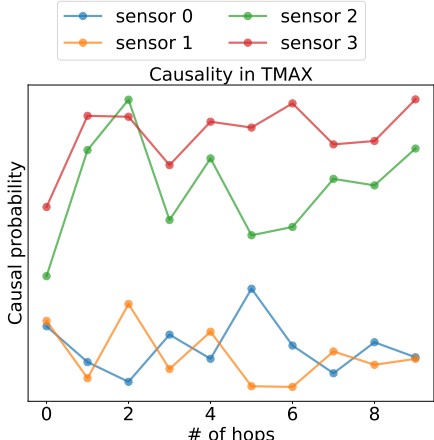

(d) Average causal probability curves vs. the number of hops from particular sensors in Eastern region (TMAX).

