# OpenReview forum: "Physics-Aware Spatiotemporal Causal Graph Network for Forecasting with Limited Data"
_TMLR — Accepted by TMLR_

### Review · Reviewer_q6C5 · 2025-04-05

**Summary Of Contributions:**

The paper proposes a novel Physics-aware Spatiotemporal Causal Graph Network (P-STCGN) designed for forecasting spatiotemporal dynamics, especially effective in scenarios with limited and noisy data. Its main contributions include:
- Introducing a method to derive causal structures from spatiotemporal data using domain-specific physics knowledge.
- Formulating a specialized neural module to learn these causal structures.
- Providing a forecasting module operating under these causal structures, enabling predictions guided by physics-informed causality.
- Extensive experimental validation demonstrating robustness and superior performance against several baselines across synthetic datasets and real-world climate data.

**Audience:**

Yes

**Broader Impact Concerns:**

The paper raises no broader impact concerns.

**Claims And Evidence:**

No

**Requested Changes:**

1. Could you please elaborate clearly on the difference between the graphs $G_s$ and $G_c$?
2. On page 3, in the bottom paragraph, could you elaborate on why the Heat equation (Eq. 1) elucidates exactly $N_c=13$ causal relations (5 temporal and 8 spatial)? A clear breakdown would help readers better understand your point.
3. Please ensure notation consistency. In Fig. 2, you denote the Spatial Encoder and Temporal Encoder with "GConv" and "FC," respectively, while in section 4.2, you refer to them as "SE" and "TE." This inconsistency may cause confusion and should be clarified.
4. In section 4.1, could you define nodes $V$ more explicitly? Earlier in the paper, each node corresponds to a single variable, but here each spatial location is defined as a node containing multiple variables (e.g., velocity vector and pressure). Clarifying precisely what constitutes a node would improve readability.
5. Please include a detailed proof for Theorem 1 to support the theoretical claims clearly.
6. In section 4.2, it would greatly help to explicitly state the training objective (loss function) of your proposed method.
7. In Theorem 1 (b), "Causality preservation," there seems to be a contradiction: initially, it states, "each variable's rate of change is solely determined by variables in its causal neighborhood," yet the subsequent equation includes an external term $g_i(t)$. Please clarify or reconcile this inconsistency.
8. In section 5.1.3, you use the notation $t$ for temperature, while earlier you use $t$ for time, causing potential confusion. Please adopt a distinct variable to represent temperature explicitly.
9. Typos to fix:
- page 4 (bottom paragraph): "node $i$" should be "node $(i,j)$".
- Equation 4: "$u_{i,i}$" should be "$u_{i,j}$".
- Page 12: "$40\\%$" should be "$60\\%$".

If the authors address all of these requested changes thoroughly, I would be pleased to recommend this paper for acceptance.

**Strengths And Weaknesses:**

Strengths:
- The paper successfully bridges physics-based inductive biases and causal inference in spatiotemporal forecasting. The use of domain-specific PDEs as causal priors is both innovative and broadly relevant across various scientific and engineering domains.
- The research addresses a well-motivated and practical challenge, demonstrating clear advantages when forecasting with limited and noisy data—a common scenario in real-world applications.
- Extensive experimental evaluations validate the effectiveness and robustness of the proposed method.

Weaknesses:
- The presentation of the paper requires improvement for better clarity and accessibility. Specific suggestions for enhancements are provided below.

---

> ### Author Response · Authors · 2025-05-08
> **Response to comments and requested changes**
>
> **Q1**: The graphs $\mathcal{G}_s$ and $\mathcal{G}_c$ are fully defined in Section 3. $\mathcal{G}_s$ is a static graph constructed from observational data, while $\mathcal{G}_c$ is a causal graph that represents the spatiotemporal relationships captured by available physics knowledge, and is constructed on a domain-specific basis (an example of which is provided in Section 4.1).
>
> **Q2**: As shown in Figure 1, green arrows indicate temporal relationships. Given 5 nodes, we have 5 "identity" temporal relationships, connecting the nodes to themselves across time steps. The blue arrows indicate spatial relationships, i.e., how a node may be influenced by its spatial neighbors through time. The number of spatial relationships of a node is equal to the number of neighbors is has. Hence, for the 5 nodes in Figure 1, we have $1 + 1 + 1 + 2 + 3 = 8$ spatial relationships.
>
> **Q3**: In Figure 2, the terms “GConv” and “FC” are intended as references to the underlying architectures (“graph convolutional layer” and “fully connected layer,” respectively) rather than aliases for those components. We have clarified this in the figure caption. Note also that the architectural details of each component can be found in Appendix B.1.
>
> **Q4**: Section 4.1 provides one particular example of how the causal graph can be constructed given available physics knowledge. The formulation in Section 3 intends to define nodes as generally corresponding to spatial features, which can include observations across many variables. However, the multi-dimensional nature of node-wise observations was implicit in the notation, which we agree leaves ambiguity. In the revised copy, we’ve refined the presentation in Section 3 to clearly indicate that node-wise observations can be feature vectors of dimension $D_v$; the $N$ spatial observations at time $t$ are referred to as $X_t \in \mathbb{R}^{N\times D_v}$.
>
> **Q5**: A detailed proof for Theorem 1 is now included in Appendix A of the revised copy.
>
> **Q6**: We have added the total loss function in Section 4.2 (Eq. 11) in the revised manuscript.
>
> **Q7**: To clarify, the term "causal neighborhood" in our formulation includes both endogenous (system) variables and exogenous influences that are explicitly modeled, such as $g_i(t)$. Thus, the expression is consistent with the stated causality-preservation principle, as all terms influencing $u_i$ are causally accounted for. We’ve refined the statement in Theorem 1 (b) to reflect this.
>
> **Q8**: In our revised copy, we’ve changed the usage from $t$ to $\mathcal{T}$ to clearly represent the temperature variable.
>
> **Q9**: Thank you for pointing these out, each typo has been corrected in the revised manuscript.

---

> > ### Comment · Reviewer_q6C5 · 2025-05-12
> > **Response to authors: All concerns have been addressed**
> >
> > All of my concerns have been satisfactorily addressed by the authors. I have no further questions. Thank you to the authors for their thoughtful responses.

---

### Review · Reviewer_R7Ye · 2025-04-07

**Summary Of Contributions:**

This paper proposes P-STCGN, a spatiotemporal causal graph network incorporating physical rules as priors. The model is a novel approach to integrating universal physical equations with data-driven causal structure learning, thus benefiting spatiotemporal forecasting under limited data. The contributions made in this paper include both methodological and applied aspects, which can be summarized as follows:
1. The proposed method for incorporating information from physical equations as a prior for causal structure learning. This idea is novel and addresses the limitation of data-driven causal discovery algorithms that ignore actual physical meaning and lack generalizability.
2. This paper designs the P-STCGN model, in which the STGC module is able to efficiently utilize physical equations such as PDEs for semi-supervised causal structure learning, while the STGF is able to utilize the learned causal associations to benefit spatio-temporal sequence prediction.
3. The extensive experiments conducted on synthetic data, benchmark data, and real-world climate data show the effectiveness of the proposed P-STCGN, even with limited training samples.

**Audience:**

Yes

**Broader Impact Concerns:**

I have no concern about the ethical implications of this paper.

**Claims And Evidence:**

Yes

**Requested Changes:**

I don't think any critical changes are required.

**Non-critical recommendation to strengthen the work:**
1. Benchmarking runtime and memory usage during training/testing against baselines.
2. Showing how the learned causal probabilities correlate with the physical variables or parameters.
3. Visualizing the learned causal structure and comparing it with the physically guided one. For example, whether the method learns the 13 causal relations in Fig. 1.

**Strengths And Weaknesses:**

**Strengths:**
1. This paper proposes an innovative framework that incorporates physical laws and causal analysis. This framework overcome the inexplainability of pure data-driven methods. Besides, this framework ensures alignment with physical rules when data collected is not informative enough.
2. The experiments show that the method proposed is robust to limited data size. With only 5% training data, the P-STCGN significantly outperforms the baseline method under different metrics.
3. The method proposed is robust to noisy or biased data. The extensive experiments conducted on synthesized or semi-synthesized datasets show that with additive noise the method still performs well and the inclusion of hidden confounder can mitigate the bias in data.
4. Theorem 1 in this paper shows that physical formulas satisfying specific regularity conditions can induce DAG, thus theoretically justifying the introduction of physical formulas.

**Weakness:**
1. As the synthetic experiment shows, incomplete physical equations lead to degraded performance, which means that this method relies on the correctness of physical rules given.
2. The method proposed includes a causal module with computational complexity of $N^2 K^2$, and the cost relies on the stationary assumption to be reduced, which limits the application of P-STCGN to non-stationary systems.

---

> ### Author Response · Authors · 2025-05-08
> **Response to comments and requested changes**
>
> (Note: items below are numbered according to list items under the *"Non-critical recommendation to strengthen the work"* section of the review.)
>
> **Q1**: In Appendix B.2, we’ve now added specific parameter counts for individual model components, detailed reports for training and inference times on our primary real-world task, and additional details regarding hardware configuration used for experiments. The reported training/inference times are measured with and without the causal component (i.e., STGF vs P-STCGN), highlighting the marginal computational costs incurred and allowing one to reason about when the performance advantage may be worth additional compute in practice.
>
> **Q2-3**: One possible clarifying point here: the causal relations found in physics equations, like those demonstrated in Figure 1, are explicitly provided to the model to use when learning to predict future time steps. The causal module (CM) is effectively encouraged to produce embeddings that reflect this structure, insofar as they aid its ability to forecast accurately. For instance, in a setting involving the heat dissipation as depicted in Figure 1, the 13 causal relations are made available to the model during training; it does not have the ability to pick up other arbitrary causal connections. Therefore any structural comparison we might employ would share the underlying causal graph, although one could inspect the strengths along the weights as dictated by the causal module.

---

> > ### Comment · Reviewer_R7Ye · 2025-05-18
> >
> > I appreciate the response from the authors which has addressed my concerns about this paper. After reading the other reviews and resposes, I do not have further questions for this paper.

---

### Review · Reviewer_kcCi · 2025-04-24

**Summary Of Contributions:**

This paper proposes P-STCGN, a novel Progressive Spatio-Temporal Channel Graph Neural Network designed to improve human action recognition from skeleton data. The architecture integrates temporal attention, progressive residual fusion, and an adaptive graph learning module that selectively models joint-wise and channel-wise correlations. The model is evaluated on benchmark datasets, showing state-of-the-art or competitive performance. The core contribution is the combination of fine-grained channel correlation learning and spatio-temporal joint modeling in a progressive, attention-driven fashion

**Audience:**

Yes

**Claims And Evidence:**

Yes

**Requested Changes:**

Compare with Recent Models: Include recent baselines for a fairer performance landscape (even just as discussion about differences and justification of why not included as proper baseline)
Efficiency Discussion: Add a paragraph on computational efficiency.
This work makes a strong technical contribution and presents a novel way to combine spatial, temporal, and channel-wise features. While performance is strong, the paper would benefit from improved clarity and better context among the latest baselines. With some polishing and added reproducibility, it has the potential to be impactful.

**Strengths And Weaknesses:**

Strengths:

Novel Architecture: The progressive fusion of spatio-temporal and channel-wise representations is original and interesting.
Channel-wise Graph Learning: Adaptive channel correlation modeling is less explored and adds depth to the understanding of motion patterns.
Strong Performance: The model achieves competitive results on benchmark datasets, reinforcing its effectiveness.
Detailed Ablation Studies: The impact of each module is well-analyzed and visualized, showing clear contributions from design choices.
Clarity in Motivation: The paper clearly justifies why modeling both spatial and channel relationships can benefit skeleton-based action recognition.

Weaknesses:

Insufficient Comparison: Missing comparison with very recent baselines like DSTA-GCN or SGN++. Would help position P-STCGN more clearly.
Efficiency Metrics Missing: No clear discussion on computational cost, parameters, or inference time, important for deployment scenarios.

---

> ### Author Response · Authors · 2025-05-08
> **Response to comments and requested changes**
>
> **Baseline comparisons**: We’ve included an additional discussion in Section 2 (related work) regarding GNN-based spatiotemporal modeling to better position our model’s contributions. Here we’ve included the mentioned DSTA-GCN [1] and SGN (assuming [2]*) models, but clarify how they differ from our proposed method and why they aren’t considered as baseline candidates. In particular:
>
> DSTA-GCN employs a fuzzy neural network to construct adjacency matrices of a spatiotemporal graph’s time slices, designed with a heavy focus on traffic forecasting. This effectively brings in an additional data-driven element for connecting pairs of time steps, but it crucially does not incorporate a causal view of the temporal process and offers no way of enforcing known physics constraints. SGN also shares some overlap in that it operates on graph-like sequences, but is fundamentally a pose/gesture recognition model. This method explicitly models joint-level dynamics and is designed around skeleton-structured data, limiting its applicability to general spatiotemporal forecasting tasks. In total, these methods have several overlapping characteristics given their graph-convolutional bases, but are ultimately examples of fairly task-specific models that make a direct baseline comparison difficult (if not impossible) on our general physics-based datasets.
>
> **Deployment/Efficiency discussion**: In Appendix B.2, we’ve now added specific parameter counts for individual model components, detailed reports for training and inference times on our primary real-world task, and additional details regarding hardware configuration used for experiments. The reported training/inference times are measured with and without the causal component (i.e., STGF vs P-STCGN), highlighting the marginal computational costs incurred and allowing one to reason about when the performance advantage may be worth additional compute in practice.
>
> [1]: Q. Zheng and Y. Zhang, "DSTAGCN: Dynamic Spatial-Temporal Adjacent Graph Convolutional Network for Traffic Forecasting," in IEEE Transactions on Big Data. 2023.
>
> [2]: Zhang, Pengfei, et al. "Semantics-guided neural networks for efficient skeleton-based human action recognition." proceedings of the IEEE/CVF conference on computer vision and pattern recognition. 2020.
>
> *SGN++ appears to be a model variant of SGN, but we couldn’t identify a specific published paper or model with this name.

---

> > ### Comment · Reviewer_kcCi · 2025-05-31
> > **Official Recommendation**
> >
> > My concens have been addresses, the discussion have been updated, computational cost, in which even that the primarly proposed causal module is significantly larger in number of parameters, its training times does not increase that large. I do not have further questions for this paper.

---

### Author Response · Authors · 2025-05-08
**Revision uploaded and individual reviewer responses submitted**

**To all reviewers**: Thank you for your valuable feedback as we improve our paper. Please find our itemized comments below and associated changes in the uploaded revision. We’re more than happy to answer any follow-up questions you may have if you find we did not sufficiently address your concerns. Thank you again for your time reviewing our work!

---

### Decision · Action_Editor_2T5z · 2025-06-22

**Recommendation:** Accept as is

**Additional Comments:**

This paper provides a new method that employs physical rules in spatiotemporal causal inference. The method is novel and the finds are supported by both theoretical and extensive empirical evidence. There were some technical concerns raised by the reviewers, but the revised version is found satisfactory by all the reviewers. All the reviewers recommend acceptance, and I am in agreement with their assessments.

**Audience:**

Yes

**Audience Explanation:**

Causality researchers working on spatial temporal data will be interested in the findings of this paper.

**Claims And Evidence:**

Yes

**Claims Explanation:**

The claims made in the paper has been verified by extensive experiments.